# Automatic Indoor as-Built Building Information Models Generation by Using Low-Cost RGB-D Sensors

**DOI:** 10.3390/s20010293

**Published:** 2020-01-04

**Authors:** Yaxin Li, Wenbin Li, Shengjun Tang, Walid Darwish, Yuling Hu, Wu Chen

**Affiliations:** 1Shenzhen Research Institute, The Hong Kong Polytechnic University, Shenzhen 518057, China; wu.chen@polyu.edu.hk; 2Department of Land Surveying and Geo-Informatics, The Hong Kong Polytechnic University, Hong Kong, China; wb.li@polyu.edu.hk (W.L.); lindsayuling.hu@polyu.edu.hk (Y.H.); 3Guangdong Key Laboratory of Urban Informatics & Shenzhen Key Laboratory of Spatial Smart Sensing and Services & Guangdong Laboratory of Artificial Intelligence and Digital Economy (SZ) & Research Institute for Smart Cities, School of Architecture and Urban Planning, Shenzhen University, Shenzhen 518052, China; shengjuntang@szu.edu.cn; 4Department of Electronic and Informatics, Faculty of Engineering, Vrije Universiteit Brussel, 1050 Brussels, Belgium; dawalid@etrovub.be; 5Geomatics Engineering Lab, Civil Engineering Department, Faculty of Engineering, Cairo University, Cairo 12613, Egypt

**Keywords:** as-built BIMs, automatic, RGB-D sensors

## Abstract

To generate indoor as-built building information models (AB BIMs) automatically and economically is a great technological challenge. Many approaches have been developed to address this problem in recent years, but it is far from being settled, particularly for the point cloud segmentation and the extraction of the relationship among different elements due to the complicated indoor environment. This is even more difficult for the low-quality point cloud generated by low-cost scanning equipment. This paper proposes an automatic as-built BIMs generation framework that transforms the noisy 3D point cloud produced by a low-cost RGB-D sensor (about 708 USD for data collection equipment, 379 USD for the Structure sensor and 329 USD for iPad) to the as-built BIMs, without any manual intervention. The experiment results show that the proposed method has competitive robustness and accuracy, compared to the high-quality Terrestrial Lidar System (TLS), with the element extraction accuracy of 100%, mean dimension reconstruction accuracy of 98.6% and mean area reconstruction accuracy of 93.6%. Also, the proposed framework makes the BIM generation workflows more efficient in both data collection and data processing. In the experiments, the time consumption of data collection for a typical room, with an area of 45–67 $m^{2}$, is reduced to 4–6 min with an RGB-D sensor from 50–60 min with TLS. The processing time to generate BIM models is about half minutes automatically, from around 10 min with a conventional semi-manual method.

## 1. Introduction

Building information models (BIMs), including as-designed BIMs (AD BIMs) and as-built BIMs (AB BIMs), are the digital representations for the whole life cycle management from design, construction to demolition [1], which are widely used in the areas of construction management [2], computer games [3], indoor navigation [4], and emergency response [5]. Different from traditional 3D models contain spatial information only, the BIMs have more information (e.g., material, functions, and topological information) and have digital representations of related information, which makes the modification and maintenance operation more practical and efficient. AD BIMs generation is the process in the design stage of facilities and becoming more and more common, while AB BIMs generation is to create models for existing facilities, which is more challenge, especially for the indoor environment considering disturbances from numerous occlusion and complex non-planar elements [6]. Furthermore, most of the existing facilities do not have the BIM document, or the BIM document is out of date due to constant renovation [7].

The standard process of generating indoor BIMs for existing buildings includes three steps [1,6,8]. The first step collects a 3D point cloud with spatial and/or color information. For the traditional surveying methods, pieces of the range-based equipment, such as the range finder, handheld 3D scanner [9], or the terrestrial laser scanner (TLS) [10], are widely used to capture 3D point clouds of facilities. Meanwhile, 2D image-based sensing technologies, such as RGB cameras, can also be used for the 3D point cloud generation with the help of structure from motion (SfM) [11,12] or simultaneous localization and mapping (SLAM) technologies [13,14,15]. The second step is the semantization of the 3D point cloud. That semantic information includes not only the spatial information, such as positions, dimensions (areas and volume), and attribute information (material and functions) of the individual construction components, but also the relationship to each other, such as dependency, topology, or joints. In the final step, well-trained and professional modelers create AB BIMs manually, referring to the semantic information. However, there are many problems with this conventional approach. Firstly, TLS is very expensive and ponderous which makes it not efficient and economical to collect the 3D point cloud, especially for the indoor environment [1]. The RGB camera is cheaper and easier to use in the indoor environment, but the 3D point cloud extraction from camera data requires a heavy computation cost, and time-consuming processing, and extra scale control. Also, the dense 3D point cloud is not always available with an RGB camera, especially in indoor environments with low lighting or sparsely textured areas [16]. Recently, some handheld 3D scanners (e.g., Zeb one and Zeb Revo), with the advantages of relatively low-cost and portable when compared with LTS, are widely used in the 3D modeling field. However, the price of them still up to several thousand dollars, which is not affordable for some commercial users. Besides, most of them require an external battery which is very heavy and not friendly for the data collection operation. Secondly, because the manual semantic processes are labor-intensive and skill-demanding, the productivity of those methods cannot keep up with the growing requirement for BIMs in the architectural/engineering/construction and facility management industry (ACE/FM).

In the past few years, researches have paid great efforts trying to find feasible solutions to reduce the cost and to enhance the efficiency of the indoor AB BIMs generation. To reduce the cost of the hardware, some researchers [16,17,18,19] proposed to apply the RGB-D technology to the indoor 3D reconstruction, as RGB-D sensors have the advantages of low cost, excellent portability and provision of synchronized color images together with dense depth maps, compared with traditional TLS and RGB camera. However, the 3D semantic information extraction from the point cloud from RGB-D sensors becomes more difficult, as the depth measurement quality of RGB-D sensors is much lower than that from TLS.

Meanwhile, some works were focused on replacing the manual process of 3D point cloud semantization with “automatic” or “semiautomatic” approaches, which can be divided into two groups: data/model-driven method [1] and deep learning-based methods.

The data-driven methods extract the semantic building information based on the features, shapes, materials, and statistics of raw measurement data, whereas model-driven methods customarily utilize predefined information or knowledge. Most of the current methods were developed by integrating both of them. Okorn et al. [20] proposed a method to extract the floor and ceiling data by using a histogram of height data with the assumption that the coordinate system of point cloud always being consistent with the real world in the height direction. The remaining points are projected onto a 2D ground plane, which is estimated from the floor data, and the line segments corresponding to walls are extracted based on the Hough transform. However, the output of this method is only the floor plans, and other elements, such as the window or door, are unavailable. Also, the line segmentation would be problematic when the dataset with too many noises. In the method developed by Wang and Cho [21], the surfaces of the elements are estimated firstly by using the region growing algorithm, and the building geometries are then extracted from unorganized point clouds by collecting the intersection lines created by two extended surfaces in vertical and horizontal. This method is practical for high-quality point clouds collected by TLS, but it would be difficult to apply to the case when the quality of point clouds is low, i.e., the region growing algorithms are useless when the wall plane is narrow, or data have strong noise. Furthermore, some methods reconstruct the properties of doors and windows as well as the relationship between them and walls by detecting the openings. Previtali et al. [22] integrated the openings detection and a shape-driver algorithm to generate the BIM model with the windows and doors at the assumption of Manhattan-World geometry. This method provides a potential solution for the relationship reconstruction of sub-element (i.e., doors and windows) and walls, but the opening detection-based algorithms require the doors/windows should being opening during data collection. Macher et al. [6] proposed a semi-automatic approach for indoor 3D reconstruction of the entire building to reduce the time consumption and errors caused by the manual operation. However, this approach also detected the window and door based on openings. Moreover, this approach may be impractical to handle the low-quality point cloud because it was designed for high-quality point cloud collected by LTS. This similar problem is existing in [23,24].

The development of deep-learning technology opens new avenues for automatic AB BIMs generation but still in its infancy. Dai et al. [25] developed one indoor scene reconstruction system called ScanNet. This system firstly produces a coarse semantic segmentation result by using deep-learning and then refines the result manually in a WebGL application. The output only includes limited attribute information rather than full digital spatial and relationship information. Another work (named IM2CAD [26]) uses only one single photo as the input and output the CAD model of the scene by using a fully convolutional network (FCN) and faster region-based convolutional neural networks (FRCN) integrated architecture. This method is innovative for the information semantization but unpractical for automatic AB BIMs generation because the process from image to CAD model would miss important details of the sense (such as small elements, and boundaries of objects), and the BIM requires a 3D model for the whole environment rather than a single scene.

Although recent development makes the generation of indoor AB BIMs more efficient and convenient, there are still many problems need to be overcome: (1) current approaches are designed for high-quality point cloud and are impotent for the low-quality dataset from low-cost sensors, i.e., RGB-D sensors; (2) traditional method is inefficient and restricted because additional manual intervention is required in semantization and digitization; (3) the extraction of digital spatial and relationship information still not been addressed, especially for the low-quality point cloud dataset.

In this paper, we aim at establishing an automatic and efficient indoor AB BIMs generation framework with low-cost RGB-D sensors to address issues about the extraction of spatial and topological information and the digitalization related information to BIM format document. Inspired by the success of deep-learning technology in the semantic segmentation areas, we modify the fully convolutional network (FCN) to extract the attribute information of the elements (e.g., wall, ceiling, floor, windows, and door) from low-quality RGB-D data (color and depth images) [27]. Furthermore, a new method to extract digital spatial information and their relationships from the low-quality point cloud is proposed through its error distribution and some empirical knowledge of the indoor environment. Finally, we implement the workflows from raw color and depth sequences in commercial software to generate BIMs automatically.

The organization of the paper is as follows: Section 2 describes the workflow as well as the detail algorithms of the system. Section 3 outlines the result of the experiment. Section 4 includes the conclusions and discussions on the future development of the system.

## 2. Automatic BIM Generation Framework

This section describes the workflows, as well as the detail algorithms of our automatic BIM generation method. As shown in Figure 1, the whole framework can be divided into three stages.

The first stage is to collect datasets and to transfer the dataset to the structure accepted by neural networks. The raw RGB images and depth images are collected by an RGB-D sensor. Then, we apply the calibration method proposed in [28] to reduce the systemic error of the RGB-D sensor. A depth information encoding method proposed by Gupta et al. [29], named HHA, is used to encode the depth information as the neural networks are weak to process the unstructured geometric information duo to the fixed grid kernel structure. Finally, the RGB images, HHA images together with the ground truth labels, are packed together as the input of the training procedure of semantic 3D reconstruction stage.

The second stage is to get the semantic 3D reconstruction result of the environment. First of all, a neural network is established for the 2D image semantic segmentation in the training procedure. This neural network is trained first with the image pairs and ground truth labels and is used to predict the semantic label of each pixel in the prediction procedure with the input of RGB images and HHA images. Then, with the input of RGB images and depth images, the 3D reconstruction is done by using the simultaneous localization and mapping (SLAM) method [30] with the output of the textured point cloud. Finally, the point cloud with semantic labels is generated by integrating the 2D semantic segmentation result and 3D reconstruction product.

The final stage is to extract the digital spatial and relationship information for the BIM generation. In this stage, the semantic point clouds are divided into different parts based on the labeled information provided in stage 2. The planes of floors and ceilings are estimated first by using the random sample consensus (RANSAC) algorithm [31]. As mentioned in Section 1, the point clouds collected by low-cost RGB-D sensors are much noisy, which makes it challenging to extract planes with conventional plane extraction methods. We propose a new “add-remove” method to overcome this problem to improve the quality of the extracted wall planes. Then, the wall planes are projected to the floor plane to get the 2D map of the walls. The start points and the endpoints of all lines are recovered using the line fitting algorithm. At the same time, the door and window point cloud are segregated into several parts which only have one element by using local descriptors [32]. The position of the elements and their relationship to walls are recovered by calculating the closest distances. Meanwhile, the sizes of the elements are calculated based on the single frame point cloud rather than all point clouds to reduce the effects of measurement noise. Finally, the BIM model is automatically generated by integrating all the segmentations with one plug-in program under the Revit platform.

### 2.1. Data Collection and Preprocessing

The hardware used to collect data in this paper is one type of RGB-D sensors named Structure sensor, which can be fitted to an iPad, an iPhone, or other mobile instruments (Figure 2). The system outputs 640 × 480 aligned RGB and depth images with the frequency up to 30 frames per second. The benefit of semantic segmentation with the depth information has been demonstrated by many studies [33,34,35]. However, the depth image from low-cost RGB-D sensors contains notable systemic errors. The calibration method [28] is first applied to improve depth measurement accuracy. The compare between the raw depth values and calibrated depth values for a wall is shown in Figure 3, and it is clearly shown the improvement of the calibrated depth as the wall should be a straight line. Additionally, as shown in Figure 4b, the occlusion or clutter, caused by the unpredictable variation of scene illumination, reflection of objects, or the out-of-range issues, makes the use of depth information more challenging. In this paper, the algorithm proposed by Levin et al. [36] is adopted to fill the occlusion region (as shown in Figure 4c), considering the filled depth is the input of the depth encoding algorithm used in the following steps. Finally, as shown in Figure 5, the depth information is encoded as HHA (horizontal disparity, height above ground, and the angle with gravity) format because the neural networks are weak to process the unstructured geometric information duo to the fixed grid kernel structure [29].

### 2.2. Semantic 3D Reconstruction

The semantic 3D reconstruction stage is divided into two parts. The first part is the semantic segmentation in 2D images based on the RGB information and depth information. The second part is the integration of 2D semantic segmentation results with RGB-D SLAM mapping based on the relationship between pixel-level semantic labels and point clouds.

For the pixel-level semantic segmentation task, the FCN [27] is accepted as the standard approach based on deep-learning, which is the first end-to-end neural network adopting the arbitrary size input and output pixel-level dense with the corresponding size [27,38,39]. However, the semantic segmentation result from FCN is not elaborate enough even with the combination of high layer coarse information and low layer excellent information. The conditional random field (CRF) [40] is one of the widely used algorithms to refine the semantic segmentation result of RGB images, but it only uses the boundary information from the color image. Meanwhile, convolutional oriented boundaries (COB) is used to extract boundary information based on the depth information [41]. Considering that the RGB-D sensors provide aligned color and depth images, we added a CRF layer and a COB layer at the end of raw two-branch FCN architecture to refine the semantic segmentation result of the neural network. The neural network used is still one end-to-end architecture with the image pairs as input and pixel-level semantic segmentation as output. In this paper, the label classes we used include door, window, floor, ceiling, and wall.

Then, we generate the semantic 3D point clouds by integrating the textured point clouds from RGB-D SLAM mapping and the 2D semantic segmentation from the neural network. First of all, the RGB-D SLAM method used by Endres et al. [42] is employed to generate the 3D point clouds from the RGB and depth information. This graph-based SLAM system calculates geometric relationships of adjacent frames based on RANSAC and iterative closest point (ICP). For each frame, the textured point cloud in the local coordinate system is generated based on the color and depth information. The textured 3D point cloud is generated by combining the point clouds of different frames with transformation matrixes from the SLAM system. Secondly, with the depth information and pixel-level semantic segmentation result, the semantic point cloud for each frame is outputted. Finally, similar to the conventional 3D reconstruction method, the semantic 3D point clouds are generated by integrating the semantic point cloud of each frame and the transformational information from the SLAM system. In practice, the overlap of adjacent frames and the incorrect 2D semantic segmentation result make label information of the 3D location ambiguous. Our system accepts to fuse the semantic segmentation results with overlap by using a Bayesian update. In practice, by using the transformation information provided by SLAM, the segmentation results can be aligned into the same coordinate system, and the overlap information is used to update the label probability distribution. The recursive update function is shown in Equation (1).
(1)$$P\left(X=\;L_{i}|C_{1,\ldots \ldots ,m}\right)=\;\frac{P\left(X=\;L_{i}|C_{1,\ldots \ldots ,m-1}\right)P\left(X=\;L_{i}|C_{m}\right)}{K},$$
where, $L_{i}$ is the prediction result of the label; $C_{k}$ is the semantic point cloud for $m^{th}$ frame; K is one constant to normalize the distribution.

### 2.3. The Transformation from Semantic 3D Reconstruction to BIM Format 3D Model

In this stage, the semantic 3D point cloud would be separated into different parts (e.g., floor, ceiling, wall, door, and window) based on the labeled information firstly. Then, the properties of the elements, as well as the bilateral relationship with each other, are extracted. Finally, the BIM format 3D models are generated based on the obtained information using the Revit platform. As shown in Figure 6, the point cloud generated from low-cost RGB-D sensors is very noisy even after the systemic error calibration, which makes most elements extraction methods not suitable for this situation. We develop one new element extraction algorithm based on the feature of the dataset as well as the empirical knowledge of the indoor environment.

#### 2.3.1. Wall Boundary Extraction

In the wall boundary extraction procedure, the wall point cloud is extracted based on the semantic label firstly, as Figure 7b shows. This point cloud always has a large number of noisy points (Figure 7b red circle areas) due to the measurement errors and aligned error of color and depth images. In this study, we remove those sparse outliers based on the distance distribution of point to neighbors in the input point cloud. The compare between the raw point cloud and the filtered result is shown in Figure 7b,c.

The first step for the wall boundary extraction is wall plane detection. As Figure 7d shows, point cloud in a different color is points for different wall planes. Traditionally, the plane extraction for a dense point cloud is based on the iteration of the RANSAC plane fitting method. For iteration i, the algorithm detects a plane $p_{i}$ from the input point cloud $\left\{P_{i}\right\}$. Those points, whose distance to the plane $p_{i}$ is less than the threshold $d_{0}$, are treated as inlier points and will be removed from $\left\{P_{i}\right\}$. The remaining point cloud is $\left\{P_{i+1}\right\}$ which is the input of the next iteration i+1. This process repeats until there are not enough points in the remaining point cloud $\left\{P_{i+1}\right\}$. This method is effective for the point cloud collected by TLS because it nearly has no overlap for the points and always in high quality. However, as Figure 8a–c shows, this method can cause an over-detection problem when the input is the low-quality point cloud collected by a low-cost RGB-D sensor. The reason for the over-detection is the value of $d_{0}$ is too small to remove all the points of plane $p_{i}$. As Figure 8c shows, some noisy points belonging to the plane $p_{i}$ are not removed, which leads to another detected plane close to plane $p_{i}.$ The parameter $d_{0}$ is used to number the inlier points for the RANSAC-based plane fitting method. In this paper, we still use $d_{0}$ for the plane fitting but use another parameter $T_{d1}$ to remove the points from $\left\{P_{i}\right\}$. This method can significantly reduce the influence the over-detection as Figure 8c–e shows.

Another problem is the over-removing of the point cloud. We remove the points close to the detected wall plane to overcome the over-detection problem. However, some of those points, which are at the joint area of walls, are useful for the detection of another plane. As Figure 9 shows, the gray point is the input point cloud, the green line is the wall plane detected by the plane fitting algorithm, and the yellow area is the region of removed points. In this case, the red rectangular area is another wall with a small size that will not be detected because the points have been removed. To address this problem, we project the detected wall plane to the floor plane to get the line of the wall in 2D mapping. The points around the endpoints of the line, the blue circle areas in Figure 9 are reserved to overcome the over-removing problem.

The detail implementation of the algorithm is presented in Algorithm 1. Where, F(.) is the plane fitting function based on RANSAC algorithm, C(.) is the function to count the point number of the point cloud, $D_{1}\left(.\right)$ is the function to calculate the distance between the points and plane, $D_{1}\left(.\right)$ is the function to calculate the minimum distance between the points and line in floor plane, Dis(.) is the function to calculate the distance between the planes, Ang(.) is the function to calculate the angle between the planes, and Proj(.) is the function to project plane to the floor plane. Additionally, Figure 10 presents an example to extract 2D wall lines from the raw point cloud. Figure 10a is the input point cloud of the walls. Figure 10b,c is the iteration of the plane detection and points removing with the method mentioned above. The iteration will terminate when the number of remaining points is less than the threshold. Then, we can get the segmented wall planes, which are set to different colors as Figure 10d. Finally, the 2D wall lines are extracted by projecting the wall planes to the floor plane and the result is showed in Figure 10e.

As shown in Figure 7h, some extracted lines are not connected because the point cloud collected not covers the whole environment, or there are still some deviations between the true value and detected results. To overcome this problem, we propose a 2D wall line connect and refine algorithm based on the vertical distances of the lines and the normal angles of the corresponding detected wall planes. Also, this method is designed with the assumption that the walls in the applied environment are perpendicular and straight. First of all, one reference line L0 are random chose from all the 2D wall lines {L}, and two vertexes of L0 are presented as the start point Points and the endpoint Pointe. Moreover, all the distance between the endpoint of L0 and the vertexes of the other lines are calculated. The line whose vertex with the minimum distance is treated as the adjacent line present as L1, and this vertex is treated as the start point of L1 and another vertex is the endpoint. Then, with line L1 as the reference line, this procedure will repeat until the found start point to be the start point of L0. In practice, as Figure 11 shows, the isolation of the lines can be divided into two groups, the first group is caused by the missing of the edges and the second group is caused by the adjacent lines are too short to get the intersection point. The classification of these two groups is based on the angle between the normal vectors of corresponding detected wall planes. For example, line Li with the plane normal vector Ni and adjacent line Li+1 with the plane normal vector Ni+1, the angle between Ni and Ni+1 could be calculated and presented as φii+1. As Figure 11 shows, if the value of φii+1 is smaller than the threshold φ0, this connection relationship is initialized as the first group. The central point of the line, whose vertexes is the endpoint of Li and start point of Li+1, are calculated. The end point of Li is adjusted to the foot of perpendicular through the central point to Li. Similarly, the start point of Li+1 is adjusted to the foot of perpendicular trough the central point to Li+1. Otherwise, if the value of φii+1 is larger than the threshold φ0, the connection relationship would be initialized as the second group. In this group, Li and Li+1 are connected by replace the end point of Li and the start point of Li+1 with the extended intersection point of two lines. Here, φ0 is equal to 45 degrees. Finally, with the height of the wall calculated from a distance between the floor plane and ceiling plane, the space boundary could be obtained. The detail implementation of the algorithm is presented in Algorithm 2, where, A(.) is the function to calculate the angle between lines, and Dv(.) is the function to calculate the close vertexes distance between lines.
**Algorithm 1** Wall Boundary extraction**Input**: Lebeled wall point cloud:{P0}Distance threshold to remove the point cloud: Td1Distance threshold to filter the plane: Td2Angle threshold to filter the plane: TaDistance threshold to reserve the point cloud: Td3Percentage threshold to end the loop: Tp**Output**: 2D Wall Lines in floor plane: L1: **initialize:** remaining point cloud:{Pr} = {P0} 2: **while**
C({Pr})< C({P0}∗ Tp)
**do**3:    Plane candidate: pc=F({Pr})4:    **if**
L is empty **then**5:       add Proj(pc) to L6:       Distance between points and plane: Dℙ= D1({Pr})7:       Distance between points and line in floor plane: DF=D2(Proj(pc), {Pr})8:       **for**
i=1, i≤ C({Pr}), i++ **do**9:          **if**
Dℙi
**<**
Td1 and DFi
**<**
Td3
**then**10:           add {Pr}i to ready to remove set {Pm}11:         **end if**12:       **end for**13:   **else**14:       **for**
j=1, j≤length(L), j++
**do**15:         Distance between planes: dp=Dis(pc, Lj)16:         Angle between planes: ap=Ang(pc, Lj)17:         **if**
dp< Td2 and ap< Ta
**then**18:           Continue19:         **end if**20:       **end for**21:       **if**
j==length(L)
**then**22:         add Proj(pc) to L23:         Distance between points and plane: Dℙ= D1({Pr})24:         Distance between points and line in floor plane: DF=D2(Proj(pc), {Pr})25:         **for**
i=1, i≤ C({Pr}), i++ **do**26:          **if**
Dℙi
**<**
Td1 and DFi
**<**
Td3
**then**27:             add {Pr}i to ready to remove set {Pm}28:          **end if**29:         **end for**30:      **end if**31:   **end if**32:   remove point from {Pr}: {Pr}= {Pr}−{Pm}33: **end while**34: **Return**: L

**Algorithm 2:** 2D Wall Lines Connection and Refining**Input**: 2D Wall Lines in floor plane: LAngle threshold: φ0**Output**: Connection vectors of 2D wall lines: L1:  **initialize**: Random Select one line L0 from L2:  Remove L0 from L, and add L0 to L3:  **while**
L is not empty **do**4:     **for**
i=1, i≤ length(L), i++ **do**5:        For distance set D: Di= Dv(L0, Li)6:     **end for**7:     Find line candidate Lc referring to the minimum value in D8:     Remove Lc from L, and add Lc to L9:  **end while**10: **for**
j=1, j≤length(L), j++
**do**11:    Calculate the angle between adjacent lines: Ang=A(Lj,Lj+1)12:     **if**
Ang< φ0
**then**13:       Extend lines to obtain intersection point14:       Update the vertexes of Lj,Lj+115:    **else**16:       Add one new line between Lj,Lj+117:    **end if**18: **end for**19: **Return:**
L

#### 2.3.2. Door and Window Extraction

The basic information of the BIM elements, such as door and windows, normally includes the position, size, and relationship with the other relative elements. In this section, we try to estimate the position and the size of doors and windows and reconstruct the relationship to the wall.

As Figure 12a,b shows, the input point cloud is segmented into several clusters based on the local descriptors [32] to make sure each cluster has only one interesting element. For each cluster, as Figure 12c shows, the point cloud is projected onto the floor plane for the 2D line fitting process. The position of the central point of the optimally fitted lines is treated as the X,Y coordinates of corresponding element. Considering the bottom of doors always aligns with the floor plane, we assign the Z value of door position as zero. For the window, the Z value of the position is calculated by projecting the point cloud to the wall plane to get the optimal fitted line. The height of the central point of the line is treated as the Z value of the window position. In this paper, as Figure 12d shows, the width and the height of the elements are estimated from the image frame rather than the global 3D point cloud because the point cloud noise makes the extraction of boundary difficult.

In the relationship reconstruction, we use the constraint that the door or the window always on the wall. Firstly, for each element, a door or window, the distances {Dis} between its position and wall planes {P}, as well as the angle {A} between the element plane and wall planes, are calculated. Then, {P} would be filtered with the condition that the value of {A} less than the threshold $\theta_{0}$, and the filtered result is $\left\{P^{f}\right\}$. Moreover, the element is subject to the wall which is in $\left\{P^{f}\right\}$ and have minimum value in {Dis}. Here, the value of $\theta_{0}$ is set to 30 degrees.

#### 2.3.3. BIM Format 3D Model Generation Based on Geometry Information

Finally, we developed a plug-in based on the Revit 2018 for the BIM format 3D model generation. As Figure 13 shows, the input is the information extracted in the last stage, and the BIM format models could be generated automatically. The first line stored a sequence of points with the format (X,Y), which represent the corner coordinate of the wall boundary in 2D projected mapping. The second line is the height of the wall calculated from the height difference between the floor plane and ceiling plane. The third line stores the information of doors with the position coordinate (X,Y), height, and width. The last line stores the information of the windows, which includes the position coordinate (X,Y,Z) as well as the height and width of the corresponding window element.

## 3. Experimental Tests and Discussion

In this section, three experiments have been done in three different classrooms at block Z of the Hong Kong Polytechnic University to test the performance of our proposed method. The operator collects the raw color and depth images by holding the hardware (showed in Figure 2) in hand and walks around the room at a given route to cover as much area as possible. The metric used to validate the performance of the algorithm includes the accuracy of element detection, the length measurement accuracy of the room dimension, and the area measurement accuracy of the main reconstructed elements. The actual values of the measurement dimension are collected manually by a range finder, and the actual values of the measurement area are calculated manually with correct dimensions. Moreover, the efficiency of the proposed method is evaluated by being compared with the TLS based method and range finder based method in the time consumption and manual load. The values of the parameters used in the test are listed in Table 1.

Figure 14 shows the detailed processes of the three experiments. In these three rooms, each room has one ceiling and one floor. Meanwhile, the room for the first experiment has two windows, two doors, and eight different walls, the second room has three windows, two doors, and eight walls, and the last room which is more complex, has three windows, two doors, and ten different walls. Figure 15 presents the BIM format 3D models generated by the proposed method as well as the ground truth manually generated by the skillful modeler.

Firstly, we validate the element extraction accuracy of the proposed, and the results are shown in Table 2. The result indicates that our proposed method extracts all the element objects, even for some small dimension walls, as shown in the red cycle area in Figure 14e.

Secondly, we compare the measured dimensions of rooms with the actual values measured by the range finder. Figure 16 shows the detail of the compare results. The first row is the measurement dimensions from our proposed method, and the second row is the true value. Also, quantitative analysis is shown in Table 3. For each room, we measure the width and length of the room, which determines the size of the room as well as the other two dimensions for evaluation. As Table 3 shows, average accuracy for three experiments is 98.6%%, 98.4%, 98.6% respectively with the maximal error for 214 mm and minimal error for 20 mm. The semantic segmentation based on deep learning can effectively extract the classes of elements in each frame, which makes the recognition more robust when compared to traditional methods.

Thirdly, considering the area is one of the most important attributions of the BIM elements, we compare the area measurement of extracted elements with the actual values, and the results are shown in Table 4. The average area measurement accuracy for three experiments is better than 91.9%, with the best 96.5% for experiment three. Meanwhile, for all three experiments, the area measurements of the walls, ceiling, and floor are in high accuracy better than 92.2%. The measured accuracies of doors and windows range from 74.7% to 96.9%, which is not as good as the other elements. The reason is that the true areas of the windows and doors are minimal, which makes the accuracy more sensitive to measurement errors.

Fourth, we test the performance of the proposed method in the “narrow” wall extraction. In this test, we treated the wall of which the length less than three meters as a “narrow” wall. There are four, four, and six narrow walls, respectively, for three experiments. As shown in Figure 17, all the narrow walls, length ranges from 319 mm to 2554 mm, are detected by the proposed algorithm. This is due to that the algorithm developed in this paper overcomes the influence of the over-detection problem significantly and prevents the removal of the point cloud of narrow walls. Also, Table 5 shows the quantitative analysis of measured narrow walls. Apart from some very narrow walls (length less than 400 mm) and individual cases, the accuracy of the most measurements is better than 80%, with the average measured accuracy 75.3%, 81.3%, and 80.5% for three experiments. There are two reasons for the narrow wall measured results are not as good as the measured room sizes. The first one is that the length of some walls is too small, which make the accuracy is sensitive to the measurement error. The second one is that the accumulation error of the SLAM system will cause a significant error at the end of the frame sequences. The closure error between the start frames and end frames makes the extraction of narrow walls more challenge especially when the point cloud is in low-quality.

Finally, we compare the time consumption of our proposed framework with the conventional TLS based method and manual surveying method. In manual surveying method, the operator should measure the dimensions of all the required elements such as the height of the room, the size of the doors, the length of the walls et al., and create the BIM format model based on collected information without the colored point cloud or 3D mesh output. The LTS used in the test is Leica ScanStation 2, which is interoperable with Lecia System 1200. Table 6 shows the information about the dataset collected by the LTS and structure sensor for three experiments. Also, a downsample operation with factor five is applied to the raw point cloud generated by the structure sensor because the frame overlapping makes the raw point cloud enormous. As Table 7 shows, using TLS to collect the dataset always costs more time and requires more manual load because of that, the setup of the equipment and the scanning process are typically time-consuming. The manual surveying would cost less time for data collection when compared to TLS based method, but the data processing work would cost more time without the point cloud as the reference. Our proposed method costs much less time than those two methods, not only in data collection but also in the data processing. In the data collection, the handheld RGB-D sensor we used does not require as much preparation work as TLS and does not need to record the measurement manually like a manual surveying method. Also, the data collection in our framework is handled by only one operator. In data processing, our method is genuinely automatic without any manual intervention and the processing time for all three cases is around 30s. With the improvements in those two aspects, the whole workflow is accelerated from about 200 min for TLS to about 17 min with our method.

## 4. Conclusions

In this paper, we proposed an automatic and efficient indoor AB BIMs generation framework by using low-cost RGB-D sensors. Firstly, we calibrate the low accuracy RGB-D sensor to increase the measurement accuracy and operation range. Then, a deep-learning-based method is used for the semantic 3D reconstruction of the indoor environment with the color and depth images pairs as the input. This method is more effective and economical when compared to traditional manual methods for segmentation. Also, we design a new procedure to transform unstructured 3D point cloud to BIM format 3D model with a low-cost RGB-D sensor.

The experiment results indicate that this method is robust and has acceptable accuracy to handle the noisy 3D point cloud with the average accuracy of 98.6% for dimension reconstruction, average accuracy of 93.6% for area reconstruction and average accuracy about 80% for the narrow wall dimension reconstruction. The time consumption is reduced from 120 min to 16.7 min for the three experiments when compared to the traditional manual TLS based method. In detail, the time requirement of data collection is reduced from 170 min to 15 min, and the time requirement of data processing is reduced from 30 min to 1.7 min. Thus, the framework proposed in this paper, which using the low-cost and portable RGB-D sensor to replace the costly TLS to collect the 3D indoor dataset, provides a potential solution for the AB BIMs generation. The next step of this research will address the issues about the extraction of the attribute (e.g., material and functions) information of the individual construction components as well as the extraction of more complex construction components (e.g., furniture and appliance). Also, we will apply the proposed framework to other sensors, such as Structure Mark II and Microsoft Azure Kinect DK, to get a better result.

## Figures and Tables

**Figure 1 sensors-20-00293-f001:**
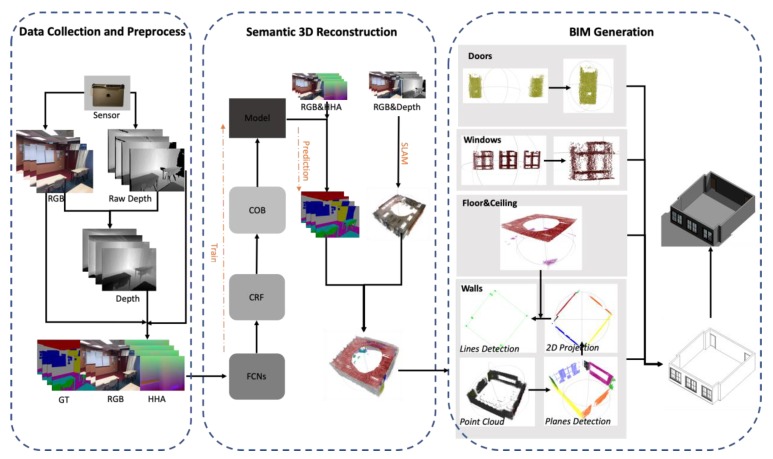
Three stages of automatic building information models (BIM) generation by using a low-cost RGB-D sensor.

**Figure 2 sensors-20-00293-f002:**
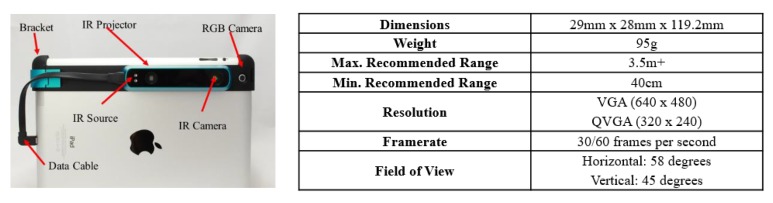
The main elements of hardware used in this paper, which includes one RGB-D camera and one iPad [37]. The total cost of the equipment is about 708 USD, 379 USD for the Structure sensor, and 329 USD for iPad.

**Figure 3 sensors-20-00293-f003:**
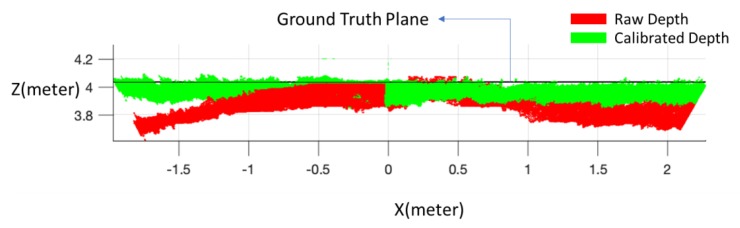
The comparison between the raw depth and calibrated depth.

**Figure 4 sensors-20-00293-f004:**
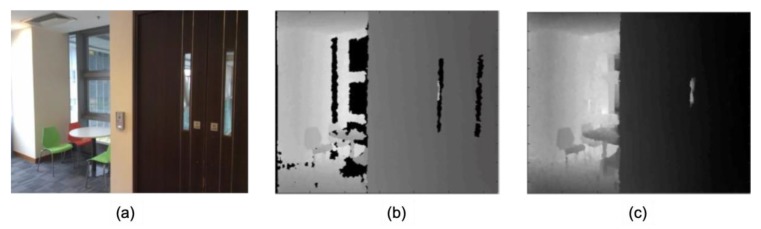
The example of (**a**) RGB image, (**b**) raw depth, and (**c**) filled depth.

**Figure 5 sensors-20-00293-f005:**
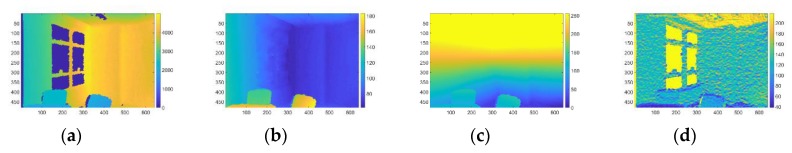
Examples of depth encoding. (**a**) Raw depth image, (**b**) the horizontal disparity, (**c**) the height above ground, (**d**) the angle with gravity.

**Figure 6 sensors-20-00293-f006:**
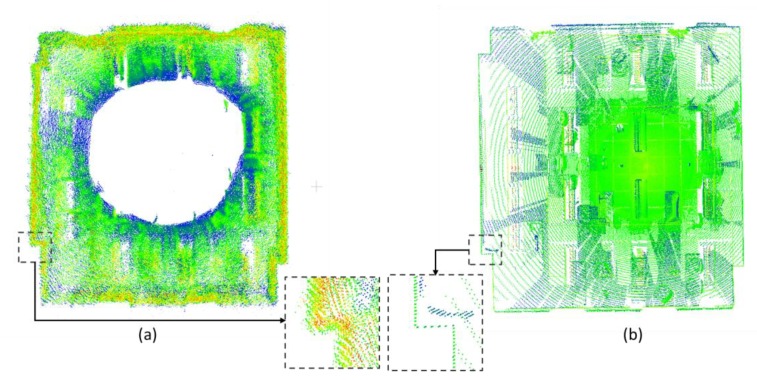
(**a**) The point cloud generated by low-cost RGB-D sensor, the empty area in the center area is caused by the data collection trajectory, (**b**) is the point cloud generated by terrestrial laser scanner (TLS).

**Figure 7 sensors-20-00293-f007:**
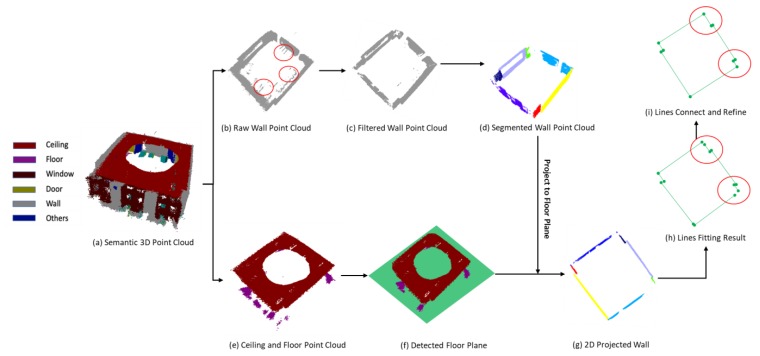
The workflow of wall boundary extraction.

**Figure 8 sensors-20-00293-f008:**
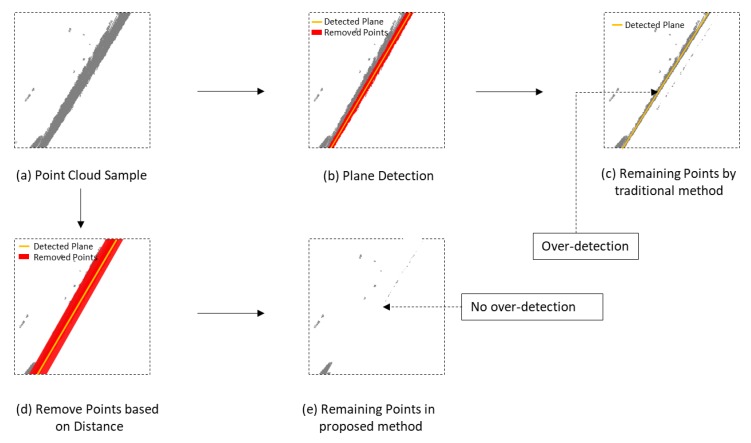
Description of the over-detection problem and the solution used in this paper.

**Figure 9 sensors-20-00293-f009:**
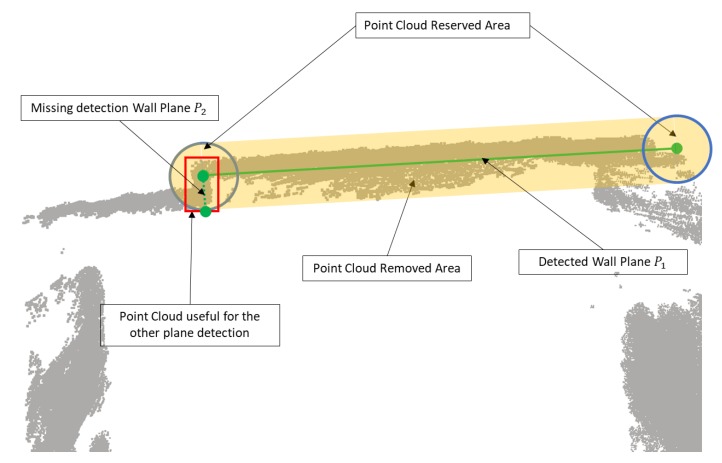
Description of the over removing problem as well as the proposed solution.

**Figure 10 sensors-20-00293-f010:**
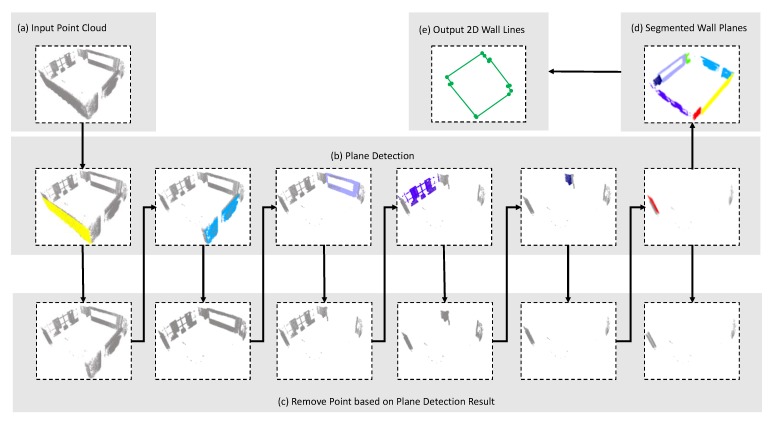
One detail example of 2D wall lines extraction.

**Figure 11 sensors-20-00293-f011:**
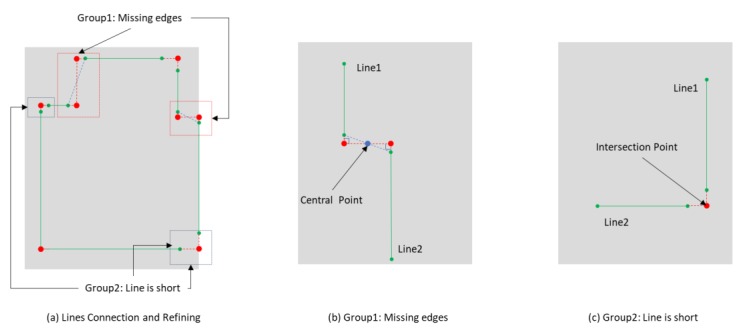
2D wall lines connection and refining.

**Figure 12 sensors-20-00293-f012:**
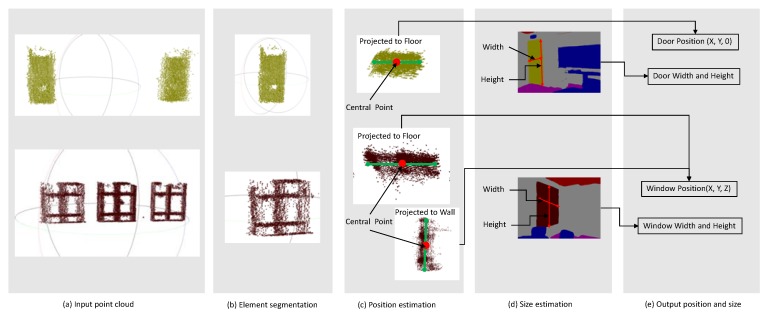
Position and size extraction of door and windows.

**Figure 13 sensors-20-00293-f013:**
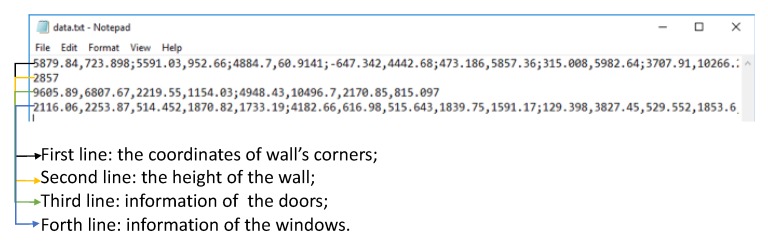
The example of the output geometry information of elements.

**Figure 14 sensors-20-00293-f014:**
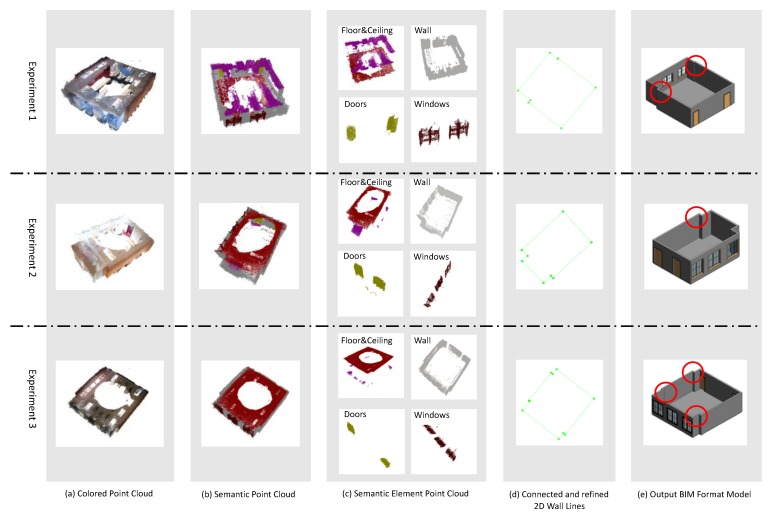
The detail processes from the raw point clouds to BIM format models.

**Figure 15 sensors-20-00293-f015:**
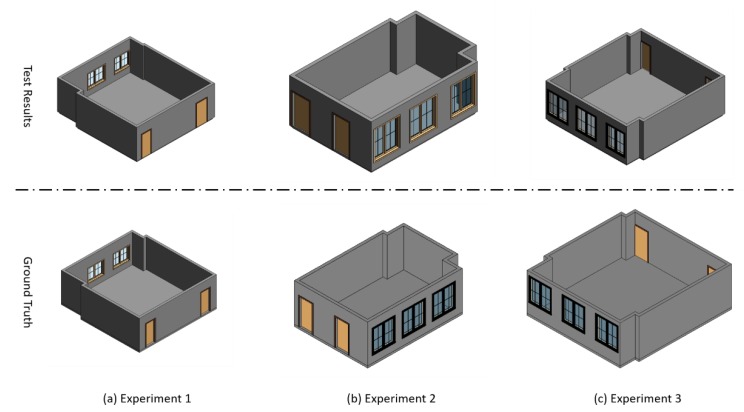
The compare of BIM format 3D models between the proposed method and ground truth.

**Figure 16 sensors-20-00293-f016:**
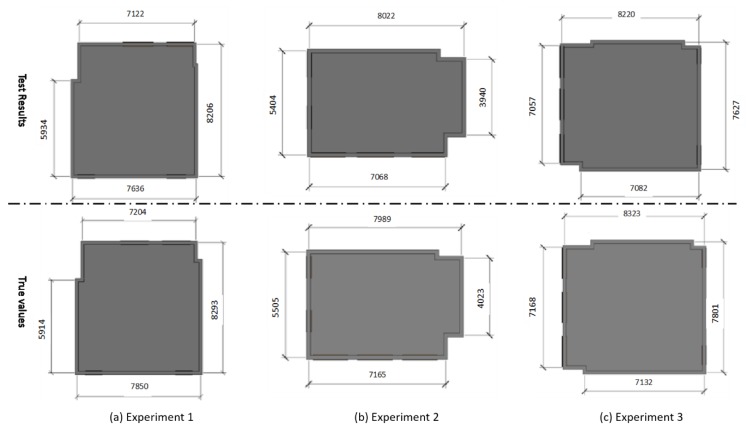
The compare between measurement dimensions of the room and the actual values collected by range finder.

**Figure 17 sensors-20-00293-f017:**
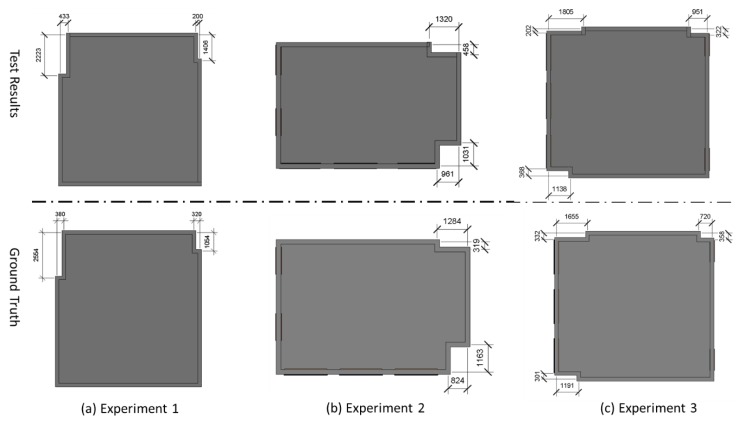
The compare between measurement length of the “narrow” walls and the actual values collected by range finder.

**Table 1 sensors-20-00293-t001:** Values of parameters used in the test.

$d_{0}\left(m\right)$	$T_{d1}\left(m\right)$	$T_{d2}\left(m\right)$	$T_{d3}\left(m\right)$	$T_{a}\left(\mathrm{degree}\right)$	$T_{p}$	$\varphi_{0}\left(\mathrm{degree}\right)$
0.1	0.25	0.25	0.25	5	0.05	45

**Table 2 sensors-20-00293-t002:** The element extraction results of the proposed method.

Experiment ID	Elements	True Number	Extracted Number	Accuracy (%)
1	Windows	2	2	100
	Doors	2	2	100
	Walls	8	8	100
	Ceiling	1	1	100
	Floor	1	1	100
2	Windows	3	3	100
	Doors	2	2	100
	Walls	8	8	100
	Ceiling	1	1	100
	Floor	1	1	100
3	Windows	3	3	100
	Doors	2	2	100
	Walls	10	10	100
	Ceiling	1	1	100
	Floor	1	1	100

**Table 3 sensors-20-00293-t003:** Quantitative analysis of measured room dimensions.

Experiment ID	Wall ID	True Length (mm)	Detected Length (mm)	Error (mm)	Accuracy (%)	Average Accuracy (%)
1	Wall1	7240	7122	118	98.4	98.6
	Wall2	5914	5934	−20	99.7	
	Wall3	7850	7636	214	97.3	
	Wall4	8293	8206	87	99.0	
2	Wall1	7989	8022	−33	99.6	98.4
	Wall2	5505	5404	101	98.2	
	Wall3	7165	7068	97	98.6	
	Wall4	4023	3904	119	97.0	
3	Wall1	8323	8220	103	98.8	98.6
	Wall2	7168	7057	111	98.5	
	Wall3	7132	7082	50	99.3	
	Wall4	7801	7627	174	97.8	

**Table 4 sensors-20-00293-t004:** Quantitative analysis of measured element areas.

Experiment ID	Elements	True Area (m^2^)	Detected Area (m^2^)	Error (m^2^)	Accuracy (%)	Average Accuracy (%)
1	Walls	88.44	86.18	2.26	97.4	92.4
	Ceiling	67.2	64.8	2.4	96.4	
	Floor	67.2	64.8	2.4	96.4	
	Doors	4.2	4.07	0.13	96.9	
	Windows	6.12	4.57	1.55	74.7	
2	Walls	67.77	65.88	1.89	97.2	91.9
	Ceiling	45.35	41.79	3.56	92.2	
	Floor	45.35	41.79	3.56	92.2	
	Doors	4.2	4.8	−0.6	85.7	
	Windows	9.18	9.87	−0.69	96.8	
3	Walls	83.55	81.65	1.9	97.7	96.5
	Ceiling	67.03	64.82	2.21	96.7	
	Floor	67.03	64.82	2.21	96.7	
	Doors	4.2	3.97	0.23	94.5	
	Windows	9.18	9.47	−0.29	96.8	

**Table 5 sensors-20-00293-t005:** Quantitative analysis of measured “narrow” walls.

Experiment ID	Narrow Wall ID	True Length (mm)	Detected Length (mm)	Error (mm)	Accuracy (%)	Average Accuracy (%)
1	Wall1	380	433	−53	86.1	75.3
	Wall2	320	200	120	62.5	
	Wall3	2554	2223	331	87.0	
	Wall4	1054	1406	−352	65.5	
2	Wall1	319	458	−139	56.4	81.3
	Wall2	824	961	−140	83.0	
	Wall3	1163	1031	132	88.7	
	Wall4	1284	1320	−36	97.2	
3	Wall1	332	202	130	60.8	80.5
	Wall2	1655	1805	−150	90.9	
	Wall3	720	951	−231	67.9	
	Wall4	358	322	36	89.9	
	Wall5	1191	1138	53	95.5	
	Wall6	301	368	−67	77.7	

**Table 6 sensors-20-00293-t006:** Information about dataset generated by TLS and the Structure sensor.

	TLS		Structure Sensor		
Experiment ID	Point Number	Station Number	Frame Number	Raw Point Number	Sampled Point Number
**1**	576,627	1	61	2,939,060	587,812
**2**	373,510	1	37	1,713,590	342,718
**3**	571,825	1	66	2,892,520	578,504

**Table 7 sensors-20-00293-t007:** Efficiency evaluation of the proposed framework.

	TLS		Range Finder		Our Proposed	
Experiment ID	Collection	Processing	Collection	Processing	Collection	Processing
1	~3600 s	~600 s	~360 s	~900 s	~300 s	34.2 s
2	~3000 s	~480 s	~300 s	~600 s	~240 s	26.7 s
3	~3600 s	~720 s	~420 s	~1020 s	~360 s	36.5 s
**Total**	200 min		60 min		16.7 min	
**Output Point Cloud**	Yes		No		Yes	
**Automatically**	No		No		Yes	
**Manual Load**	Much		Median		Little	

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
