# Peer review of "Automatic Indoor as-Built Building Information Models Generation by Using Low-Cost RGB-D Sensors"

_sensors, 2020, doi:10.3390/s20010293_

Round 1

Reviewer 1 Report

This paper describes a software system that automatically generates as-built building information models from RGB-D sequences taken by low-cost sensors.

First of all, I am concerned with the Structure Sensor that the authors used for their experiments. The sensor that was released quite while ago appears to be out of date, and even the Structure Sensor has a newer version (mark II) with increased, 1280x960 depth resolution. While there still exist mobile RGB-D sensors with, say, 640x480 pixels, I am not sure if it is timely to develop a method aimed at working with rather old devices. It might be a good idea to experiment the proposed system with a newer RGB-D sensor. The Microsoft Azure Kinect DK sensor that provides with better input data might be a good candidate.

Secondly, the authors presented a well-designed pipeline for handling the  input RGB-D sequence. However, most of techniques that comprise the system are already known. Their contributions, I think, exist in handling the noisy, error-prone point data and estimating the their boundaries, whose effectiveness is hard to prove with three example data. While they presented the accuracy measures that look fine in general, I wonder if the details like the narrow walls have been reconstructed correctly. (No discussions were found in the paper) If not, and if this is due to the low resolution and poor accuracy of the mobile sensor, I would recommend the authors to use a newer version of mobile RGB-D sensors that are easily available these days.

Again, I do not see any reason to rely on the old machinery.

Reviewer 2 Report

This paper describes a method to generate BIMs from data obtained with a low-cost sensor. It combines semantic segmentation from images with different point cloud processing methods. The work presents very good results, the introduction is very well developed and the methodology is coherent and well explained, specifically, Figure 9 exemplifies each iteration very well.. However, a BIM is much more than detecting walls and floors. How is the BIM generated here different from a 3D model? The paper has some aspects to correct:

Although the statements regarding the low-cost sensor are true, there are low-cost portable laser scanners (zeb revo, for example), what advantages would those rgbd have over them?

There are numerous works on the extraction of 3D models of buildings from point clouds, even of entire plants, but the authors do not mention any of them. This should be corrected

Line 53: "Meanwhile, 2D image-based... " It needs a reference.

Line 74, 94: LTS? or TLS?

Line 80: data/model-driven method is the same than heuristic method?

Line 122-196: The modification of the FCN is mentioned but no indication is given of how it is modified or from what work it is modified.

Line 170: even being a low cost sensor, if an iPad or iPhone is needed, this would not increase the price until be similar to other better devices? For example Hololens?

Line 180: Given the importance of Levin et al.[29] in the method, a brief description should be made.

Figure 5a: Why is there an empty zone in the center? Is it because of the way the data is acquired? Can't that zone be acquired through an acquisition with trajectory?

In results, the values of the parameters (d0...) must be indicated.

In algorithims, functions are not considered inputs in the pseudocode.

The comparison with other methods of acquisition is fundamental, it should be exposed something else of equipments that were made as acquisitions and the number of points / images generated with each method, this info is very related to processing times. As well as information of the inofmatic equipment used in processing.

A disussion is missing.

Round 2

Reviewer 1 Report

As I still think the peformance of the presented algorithm may improve markedly simply by using a newer mobile sensor that is easily available, the authors seemed to have done their best with respect to the old sensor.  I would expect to see how the presented method needs to evolve for the mobile sensor with the enhanced sensing capability.

Reviewer 2 Report

The authors have made some very good modifications. Their answers have defended their point of view very convincingly and well presented. The paper has been expanded and I consider it suitable for publication in its present form.